# A Novel Measure of Political Risk and Foreign Direct Investment Inflows

Pavel Jeutang [1] and Kwabena Kesse [2,*]

1   College of Business Administration, University of Nebraska at Omaha, Omaha, NE 68182, USA;
    njeutang@unomaha.edu
2   The Business School, University of Colorado Denver, Denver, CO 80204, USA
*   Correspondence: Kwabena.Kesse@ucdenver.edu; Tel.: +1-303-315-8482

**Abstract:** This paper proposes a novel measure of political risk that confirms some of the findings documented in the Foreign Direct Investments (FDI) literature. Particularly, we confirm the positive relationship between political stability and its components on FDI inflows, and the moderating effect of natural resources on this relationship. The proposed political risk measure contains relevant, unique and incremental information not observed in the literature. For example, although this measure is highly correlated with the political risk rating of the International Country Risk Guide (ICRG), it contains unique information that explains FDI inflows beyond what is explained by the ICRG rating. A link to the database for our political risk rating for 150 countries covering 2000 to 2015 has been provided.

**Keywords:** foreign direct investments; political risk; country risk; natural resources

## 1. Introduction

There is an extant literature that examines the relationship between Foreign Direct Investments (FDI) flows and political risk albeit with conflicting results. For example, North and Weingast (1989) and Li (2009) show that democracy has a positive effect on FDI, whereas Li and Resnick (2003) argue that autocratic governments can have a positive impact on FDI. This article re-examines the relationship between FDI and political risk by proposing a novel and robust measure of political risk (political stability index). Political risk can be a nebulous, multi-faceted and difficult to measure concept and this poses major challenges for using political risk in any empirical study. Thus, most scholars have either used some proxy variable(s) to capture some aspect(s) of political risk while others have relied on some index that encapsulates various aspects of political risk in assessing its impact on attracting FDI. Results are sometimes conflicting because the proxies used usually measure different aspects of political risk. This paper uses a novel, readily available measure of political risk that attempts to encapsulate political risk in all its various dimensions.

Perhaps the most well-known and often-used political risk index in FDI studies in particular and in the finance and economic literature in general (e.g., Loree and Guisinger (1995); Busse and Hefeker (2007); Ali et al. (2010); Hayakawa et al. (2013)), is the political risk rating of the International Country Risk Guide (ICRG) published by the PRS Group. We show that the novel political risk measure proposed in this paper (referred to as political stability index) contains unique and incremental information that is able to explain net FDI flows beyond what is explained by the ICRG rating. Pooled OLS with the political stability index as the dependent variable and the political risk rating of the ICRG as the independent variable has an R-square of 0.646 (a correlation coefficient between the two variables of 0.804). Hence, there is substantial common information between the two variables. We

do, however, find that the unique information contained in the political stability index proposed in this paper, which is unexplained by the ICRG's political risk rating, explains net FDI inflows in our sample. This unique information is simply the residuals obtained from estimating the pooled OLS regressions of our political stability index on the ICRG's political risk rating.

We confirm a positive relationship between net FDI inflows and our political stability index. As an illustration, a Sub-Saharan African country, such as Equatorial Guinea, could increase net FDI inflows by approximately 3.3% by improving their political stability score to the level of Mauritius. This relationship is statistically significant at the 1% level and economically significant. Interestingly, although both our political stability index and the ICRG's political risk variable are able to explain net FDI inflows on their own, when both variables are used in our regression model, the political stability index remains significant at the 1% level, whereas that of the ICRG drops in significance to the 5% significance level.

Additionally, we test whether political risk matters more for certain kinds of FDI than for others. Asiedu and Lien (2011) argue that the presence of natural resources moderates the relationship between FDI and democracy. They show that democracy promotes FDI if and only if the share of natural resources (minerals and oil as a share of total exports) is less than some critical value, and that a high level of democratization may actually be bad for attracting FDI to predominantly natural resource-exporting countries. A plausible explanation is that for countries that attract very large amounts of natural resource-seeking FDI, these FDI projects are tightly controlled by the government since these resources are strategically, politically and financially important to the host countries. They argue that in such countries, having close ties with the government may be the difference between a multinational corporation gaining access to natural resources or not, and such relationships are easier to foster in autocratic regimes.

It is also noteworthy that countries, such as Nigeria, Saudi Arabia, Venezuela, Iraq and many others that have over 70% of their exports being from the extractive industry, have all struggled with one form of political instability or another. For example, Nigeria has oscillated between repressive military governments, conflict and democracy whilst Iraq has been involved in several large-scale wars. Yet, these disturbances have not deterred multinationals in the extractive industry from setting up shop in these countries. We therefore test whether the presence of natural resources has a moderating effect on the relationship between FDI inflows and our political stability index. Our results align with the findings of Asiedu and Lien (2011). We do see that whereas the presence of natural resources within a country may attract FDI, their presence reduces the positive effect of a favorable political stability rating on net FDI inflows.

Finally, using a dynamic panel data setting, Busse and Hefeker (2007) find the three most important political risk factors that matter for the investment decisions of multinationals to be the stability of the government and its ability to carry out its policies and stay in office, the strength and impartiality of the legal system and the degree of ethnic tensions. Even though it is not surprising that the degree of ethnic fractionalization affects the investment decisions of multinationals[1], we find it surprising that it is one of the most important political risk considerations given the paucity of coverage it has received in the FDI literature compared to the other determinants of political risk[2].

An overview of the political economy literature suggests the relationship between FDI and ethnic tensions or fractionalization may be far from clear-cut. There are arguments for both high and low degrees of ethnic fractionalization being precursors to political instability and hence, being detrimental for FDI. Collier et al. (2003) classify societies as polarized (with two equal groups) or diverse (with many ethnic groups). They contend that the risk of rebellion increases by 50% if there is a dominant ethnic group (the largest ethnic group forms an absolute majority) and point out that this is the case with approximately half of developing countries. Similarly, Montalvo and Reynal-Querol (2002) also find the risk of civil war to be six times higher in polarized societies than in non-polarized societies. These descriptions seem to characterize countries such as Rwanda and Burundi, which, although

they have low levels of ethnic fractionalization (two major ethnic groups—Hutus and Tutsis), they are highly polarized (Hutus make up approximately 80% of the population, but Tutsis dominated politically in post-colonial times). This polarity eventually led to the Rwandan genocide and has led to several cases of instability in Burundi. On the other hand, others (Gellner (1991); Vanhanen (1999); Sambanis (2001)) posit ethnically diverse societies have higher risks of civil war, probably due to the presence of too many competing interests. In fact, Collier et al. (2003)[3] also suggest that while substantial ethnic and religious diversity reduces the likelihood of a civil war, such societies may also have less harmony compared to homogenous societies[4]. It is important to note that ethnic and cultural tensions are especially critical in natural resource-rich developing countries. Such societies are usually characterized by high inequalities and most tensions may be over which group should benefit more from the natural resources.

It is obvious from the debate in the political economy literature that it is unclear whether high or low levels of ethnocultural fractionalization pose the greater risk to political stability. Because arguments can be made for both high and low levels of ethnic fractionalization being conducive (or not) for political stability and thus attracting FDI, we test this relationship by testing for nonlinearity in the FDI-ethnocultural fractionalization relationship. We find that a nonlinear concave relationship may exist between net FDI flows and the degree of ethnocultural fractionalization, especially for natural-resource-exporting countries.

This study contributes to the literature by introducing a novel, comprehensive, readily available measure of political risk, which together with its components can be used to explain macroeconomic phenomena. This paper also complements prior empirical research that shows political risk and some of its components are determinants of FDI flows. We also complement the work of Asiedu and Lien (2011) by showing that the presence of natural resources in a country to an extent negates the positive relationship between FDI inflows and a favorable political risk rating as a whole. Asiedu and Lien (2011) focus on democracy, which is just a subcomponent of political stability.

The remainder of the paper is organized as follows: Section 2 reviews the relevant literature, Section 3 describes the data and variables, Section 4 describes the methodology and presents the empirical results and Section 5 concludes.

## 2. Related Literature

The first group of studies in this literature are those that look at some aspect of political risk and how it influences net FDI flows. For example, Asiedu and Lien (2011) study the impact of democracy on FDI and find that the relationship is different depending on whether a country is classified as natural resource exporting or not. They find that for non-natural resource-exporting countries an increase in democratization increases FDI, whereas for natural resource-exporting countries increasing democratization actually reduces FDI. Other authors focus on democracy and arrive at opposing conclusions. Whereas some (North and Weingast (1989); and Li (2009)) argue that democracy may have a positive impact on FDI[5], others argue that multinationals may in fact prefer to invest in autocratic regimes[6] (Li and Resnick (2003)). Asiedu and Lien (2011)'s results possibly shed some light on the opposing positive and negative effects of democracy on FDI. The other political risk proxy that has been explored is corruption. Asiedu (2006) uses corruption, as well as three different proxies for political stability[7] and finds a negative relationship between these political risk variables and FDI. Habib and Zurawicki (2002) and Wei (2000) also explore the impact of corruption in attracting FDI inflows and do find that corruption has an adverse effect. Finally, Gastanaga et al. (1998) use nationalization risk as a proxy for political risk and find a negative relationship with FDI.

The second strand of the literature use a political risk index or the subcomponents of a political risk index to explore its relationship with FDI. For example, Edwards (1990) uses a degree of political instability index from Cukierman et al. (1989). This index is based on the probability of government change in a given year and the degree of political

polarization and violence and is constructed as the sum of the yearly frequency of political assassinations, violent riots, protests, political attacks and politically motivated strikes. Edwards (1990) finds that reduced political instability and polarization will tend to increase FDI. A host of papers use the political risk rating from the International Country Risk Guide (ICRG) (or one or several of its 12 subcomponents), which is provided by the PRS Group. Some of the papers that relate FDI to the ICRG political risk data are Loree and Guisinger (1995), Busse and Hefeker (2007), Ali et al. (2010), Hayakawa et al. (2013), and Akbar and Khan (2013). They seem to find that different aspects of political risk matter whereas others do not. For example, Akbar and Khan (2013) find that political risk only matters for lower middle-income countries. They find that lack of government stability, socioeconomic conditions, internal conflict, religion in politics, lack of law and order, ethnic tensions, lack of democratic accountability and bureaucratic quality all adversely affect FDI inflows. Hayakawa et al. (2013) find that the political risk components that matter for FDI are socioeconomic conditions, investment profile and external conflict, whereas Ali et al. (2010) argue that the most significant institutional aspects that drive FDI are linked to proprietary rights, the rule of law and expropriation risk.

Even though the majority of studies find that political risk and institutions, however defined or measured, matter in one way or the other for FDI, some studies do not agree. For example, Asiedu (2002) finds that neither political risk nor expropriation risk matters for FDI, whereas Asiedu and Lien (2011) find that democracy matters for FDI if and only if the value of the share of minerals and oils in the total exports of a country is less than some critical value. In fact, they find 22 countries where an increase in democratization may reduce FDI. Finally, Noorbakhsh et al. (2001) fail to establish a link between democracy, political risk and FDI.

### 3. Data and Variables

The political stability index draws inspiration from Karolyi (2015) who uses principal component analysis (PCA) to estimate a political stability score. The index comprises seven sub-components that capture different aspects of political risk, namely: conflict and violence, corruption, ethnocultural fragilities, political constraints, quality of political governance, rights and freedom and welfare and socioeconomic conditions. The Appendix A provides more information about the underlying variables of each of the subcomponents.

To generate our political stability index, PCA is first performed on the underlying variables of each of the subcomponents of the political stability index one at a time and the principal component (PC) scores whose eigenvalues are greater than or equal to one are retained. Next, the retained PC scores are weighted by the relative importance of their respective eigenvalues and then summed up to generate a "raw index" for that subcomponent of political stability. Mathematically, the subcomponent raw index is obtained as follows:

$$Sub\ index\ score = \sum_{i=1}^{n} w_i PC_i.$$
$$w_i = \frac{E_i}{\sum_{i=1}^{n} E_i},\ where\ E_i \geq 1,\ E_i\ is\ eigenvalue.$$

These raw scores are then normalized to lie between zero and one such that a higher number implies more stability or that a country is doing well on that aspect of political risk, except for ethnocultural fractionalization, for which a high or low score may not necessarily be good or bad. For ethnocultural fractionalization, a zero means low fractionalization (a homogeneous society) and a one means high fractionalization (a heterogeneous society). At this point, we have seven sub-indices. Imputation techniques are used to fill missing data for some of the underlying variables. Missing data are generally imputed taking the closest available value. The countries used generally have a high availability of data.

To generate the political stability index, the same approach employed to generate the subcomponent indices is employed, but the underlying variables that go into this PCA are the seven subcomponent indices earlier obtained. At this stage, there is no missing data.

Our analysis is based on a sample of 137 countries (developed and developing countries) from 2000 to 2015. We begin our analysis in 2000 because that is the first year for which we have political risk data. The political stability index and each of its seven subcomponents are normalized to lie between zero and one such that a higher number implies more stability. We therefore expect positive coefficients on political stability and its subcomponents when examining the FDI-political stability relationship, except for ethnocultural fractionalization, of which we are unsure at this stage. As argued in the introduction, justification can be made for both a positive and negative relationship between FDI and ethnocultural fractionalization from the political economy literature.

The dependent variable in this study is net inflows of foreign direct investment as a percentage of GDP (FDI) collected from the World Bank's World Development Indicators, as well as the following control variables:

- DomCrdbyFin: domestic credit provided by the financial sector as a percentage of GDP;
- LogCellphones: natural log of mobile cellular subscriptions per 100 people;
- CapForm: gross fixed capital formation as a percentage of GDP;
- Trade: the sum of exports of goods and services (as a percentage of GDP) and imports of goods and services (as a percentage of GDP);
- LogGDP_PerCap10: the natural log of GDP per capita in constant 2010 US dollars;
- Infl_deflator: the annual GDP deflator in percentage;
- NatAgric: the sum of fuel exports, agricultural raw materials exports and ores and metals exports as a percentage of merchandise exports of a country. A country is classified as natural resource exporting if this variable is greater than or equal to 50%.

The complete sample for which we have data for all variables comprises 137 countries (see Appendix A for the full list of countries used in this study). The descriptive statistics are reported in Table 1.

Table 1 suggests that compared to non-natural resource-exporting countries, natural resource-exporting countries on average attract lower FDI, are prone to have more corruption, less rights and freedom, fewer political constraints, prone to more conflict and violence, more socioeconomic instability, poorer quality of political governance and have lower political stability scores in general.

Table 2 reports the correlation coefficients between the variables used in this study. Most of the subcomponents of the political stability index are highly correlated; ethnocultural fractionalization has the lowest correlation coefficients with the other subcomponents.

Our political stability index and its subcomponents tend to be positively correlated with FDI and with per capita GDP and negatively correlated with the volume of natural resource exports. This preliminary evidence suggests that political stability matters for net FDI flows, but this relationship may be less important for natural resource-exporting countries. This would align with the findings of Asiedu and Lien (2011) who find a similar dynamic in the relationship between net FDI flows and democracy.

The patterns about our political stability index and natural resource-exporting countries observed in Tables 1 and 2 also support beliefs and frustrations long-held in the international development community. For example, Stewart Patrick explains in a 2012 Council on Foreign Relations article (Exorcising the Resource Curse: Some Innovative Ideas) that in resource-rich countries, easy resource revenues contribute to a lack of accountability between governments and citizens. This is because these revenues eliminate the incentives for governments to tax other productive activity in the country, and as a result, makes them less compelled to deliver social services effectively. He argues elites and regime supporters enrich themselves from these revenues and stifle political reform.

**Table 1.** Summary Statistics.

| Variables | All Countries | | Non Natural Resource Exporting Countries | | Natural Resource Exporting Countries | |
|---|---|---|---|---|---|---|
| | **Mean** | **Std. Dev.** | **Mean** | **Std. Dev.** | **Mean** | **Std. Dev.** |
| FDI | 4.923 | 10.324 | 5.148 | 11.385 | 4.316 | 6.641 |
| PolStability | 0.515 | 0.258 | 0.563 | 0.257 | 0.389 | 0.212 |
| Corruption | 0.397 | 0.269 | 0.433 | 0.272 | 0.303 | 0.235 |
| EthnoCultural | 0.523 | 0.252 | 0.525 | 0.244 | 0.519 | 0.273 |
| RightsFreedom | 0.583 | 0.256 | 0.630 | 0.249 | 0.460 | 0.233 |
| PolConstraints | 0.507 | 0.265 | 0.556 | 0.259 | 0.378 | 0.233 |
| ConflictViol | 0.619 | 0.231 | 0.655 | 0.230 | 0.522 | 0.203 |
| WelfareSocioEco | 0.576 | 0.265 | 0.598 | 0.257 | 0.519 | 0.276 |
| GvceQuality | 0.474 | 0.251 | 0.516 | 0.252 | 0.365 | 0.212 |
| DomCrdtoPriv | 49.902 | 45.630 | 57.099 | 48.868 | 31.114 | 28.231 |
| LogCellphones | 3.644 | 1.515 | 3.761 | 1.389 | 3.338 | 1.766 |
| CapForm | 22.685 | 8.336 | 23.009 | 8.436 | 21.790 | 7.995 |
| Trade | 88.092 | 50.457 | 92.437 | 55.077 | 76.494 | 32.590 |
| LogGDP_PerCap10 | 8.452 | 1.563 | 8.539 | 1.570 | 8.222 | 1.520 |
| Infl_deflator | 7.129 | 14.415 | 6.256 | 10.029 | 9.430 | 21.992 |
| NatRscWagric | 31.053 | 28.806 | - | - | - | - |

This table provides summary statistics of variables used in the study from 2000 to 2015. *FDI* is net foreign direct investment inflow as a percentage of GDP. *DomCrdtoPriv* is domestic credit to private sector as a percentage of GDP. *LogCellphones* is the natural log of mobile cellular subscriptions per 100 people. *CapForm* is gross fixed capital formation as a percentage of GDP. *Trade* is the sum of exports and import of goods and services as a percentage of GDP. *LogGDP_PerCap10* is the natural log of GDP per capita in constant 2010 US dollars. *Infl_deflator* is the inflation, GDP deflator (annual %). *NatRscWagric* is a natural resource variable, which is a country's sum of fuel exports, agricultural raw materials exports, and ores and metals as a percentage of merchandise exports. The aforementioned variables are from the World Development Indicators published by the World Bank. *PolStability* is the political stability index proposed in this paper with the following seven components. *Corruption* is the corruption component, which measures the extent of corruption in a political system. *EthnoCultural* is the ethno-cultural fractionalization component, which focuses on degree of ethnic, linguistic, religious and other cultural fractionalization. *RightsFreedom* is the rights and freedom component, which focuses on the availability and protection of human rights, political rights, and civil liberties. *PolConstraints* is the political constraints component, which focuses on the presence and effectiveness of checks and balances as well as the independence of the branches of government within a political system. *ConflictViol* is the conflict and violence component, which focuses on: (1) the existence of (2) the likelihood of (3) and the residual effect of past conflict and violence in a country (both internally and externally) and in surrounding countries. *WelfareSocioEco* is the welfare and socioeconomic conditions component that is based on the interaction of living standards, poverty levels, income disparities, and availability and access to social welfare programs. *GvceQuality* is the quality of political governance component, which reflects the people's perception of how confident they are in their government's capacity to provide quality public and civil services, the ability of the government to formulate and implement policies that can foster private sector growth, and confidence in their government guaranteeing the rule of law. *PolStability* and its seven components are all normalized to lie between zero and one such that a higher number implies more political stability. Natural resource-exporting countries are those whose time series average of *NatRscWagric* is greater than or equal to 50% and those with less than 50% are classified as non-natural resource-exporting countries.

Stewart Patrick goes on to further explain " … the very presence of oil and gas resources within developing countries exacerbates the risk of violent conflict. The list of civil conflicts fought at least in part for control of oil and gas resources is long. A partial list would include Nigeria, Angola, Burma, Papua New Guinea (Bougainville), Chad, Pakistan (Balochistan), and of course Sudan. Studies confirm that the risk of civil war greatly increases when countries depend on the export of primary commodities, particularly fossil fuels."

In the next section, we develop and test hypotheses regarding our political stability index and net FDI inflows.

**Table 2.** Correlation Matrix.

| | FDI | PolStability | Corruption | EthnoCultural | RightsFreedom | PolConstraints | ConflictViol | WelfareSocioEco | GvceQuality | DomCrdtoPriv | LogCellphones | CapForm | Trade | LogGDP_PerCap10 | Infl_deflator | NatRscWagric |
|---|---|---|---|---|---|---|---|---|---|---|---|---|---|---|---|---|
| FDI | 1.000 | | | | | | | | | | | | | | | |
| PolStability | 0.070 | 1.000 | | | | | | | | | | | | | | |
| Corruption | 0.085 | 0.834 | 1.000 | | | | | | | | | | | | | |
| EthnoCultural | −0.015 | −0.040 | −0.028 | 1.000 | | | | | | | | | | | | |
| RightsFreedom | 0.076 | 0.960 | 0.710 | −0.025 | 1.000 | | | | | | | | | | | |
| PolConstraints | 0.040 | 0.963 | 0.708 | −0.040 | 0.940 | 1.000 | | | | | | | | | | |
| ConflictViol | 0.089 | 0.829 | 0.782 | −0.084 | 0.756 | 0.705 | 1.000 | | | | | | | | | |
| WelfareSocioEco | 0.041 | 0.676 | 0.721 | −0.186 | 0.524 | 0.549 | 0.702 | 1.000 | | | | | | | | |
| GvceQuality | 0.086 | 0.872 | 0.930 | −0.052 | 0.749 | 0.745 | 0.812 | 0.796 | 1.000 | | | | | | | |
| DomCrdtoPriv | 0.065 | 0.628 | 0.708 | −0.016 | 0.518 | 0.521 | 0.557 | 0.651 | 0.748 | 1.000 | | | | | | |
| LogCellphones | 0.069 | 0.394 | 0.440 | −0.081 | 0.335 | 0.315 | 0.433 | 0.430 | 0.417 | 0.437 | 1.000 | | | | | |
| CapForm | 0.183 | 0.008 | 0.034 | −0.078 | −0.016 | −0.027 | 0.098 | 0.120 | 0.050 | 0.057 | 0.183 | 1.000 | | | | |
| Trade | 0.340 | 0.100 | 0.216 | 0.029 | 0.057 | −0.011 | 0.310 | 0.214 | 0.226 | 0.132 | 0.180 | 0.225 | 1.000 | | | |
| LogGDP_PerCap10 | 0.058 | 0.708 | 0.790 | −0.149 | 0.564 | 0.576 | 0.745 | 0.904 | 0.815 | 0.686 | 0.573 | 0.109 | 0.253 | 1.000 | | |
| Infl_deflator | −0.004 | −0.215 | −0.207 | 0.032 | −0.202 | −0.186 | −0.202 | −0.127 | −0.240 | −0.224 | −0.228 | −0.060 | −0.009 | −0.156 | 1.000 | |
| NatRscWagric | −0.038 | −0.413 | −0.304 | 0.055 | −0.410 | −0.414 | −0.311 | −0.201 | −0.366 | −0.308 | −0.134 | −0.063 | −0.152 | −0.146 | 0.110 | 1.000 |

This table provides the correlation coefficient between the variables used in this study from 2000 to 2015. *FDI* is net foreign direct investment as a percentage of GDP. *DomCrdtoPriv* is domestic credit to private sector as a percentage of GDP. *LogCellphones* is the natural log of mobile cellular subscriptions per 100 people. *CapForm* is gross fixed capital formation as a percentage of GDP. *Trade* is the sum of exports and import of goods and services as a percentage of GDP. *LogGDP_PerCap10* is the natural log of GDP per capita in constant 2010 US dollars. *Infl_deflator* is the inflation, GDP deflator (annual %). *NatRscWagric* is a natural resource variable which is a country's sum of fuel exports, agricultural raw materials exports, and ores and metals as a percentage of merchandise exports. The aforementioned variables are from the World Development Indicators published by the World Bank. *PolStability* is the political stability index proposed in this paper with the following seven components. *Corruption* is the corruption component, which measures the extent of corruption in a political system. *EthnoCultural* is the ethno-cultural fractionalization component, which focuses on degree of ethnic, linguistic, religious and other cultural fractionalization. *RightsFreedom* is the rights and freedom component, which focuses on the availability and protection of human rights, political rights, and civil liberties. *PolConstraints* is the political constraints component, which focuses on the presence and effectiveness of checks and balances as well as the independence of the branches of government within a political system. *ConflictViol* is the conflict and violence component, which focuses on: (1) the existence of (2) the likelihood of (3) and the residual effect of past conflict and violence in a country (both internally and externally) and in surrounding countries. *WelfareSocioEco* is the welfare and socioeconomic conditions component that is based on the interaction of living standards, poverty levels, income disparities, and availability and access to social welfare programs. *GvceQuality* is the quality of political governance component, which reflects the people's perception of how confident they are in their government's capacity to provide quality public and civil services, the ability of the government to formulate and implement policies that can foster private sector growth, and confidence in their government guaranteeing the rule of law. *PolStability* and its seven components are all normalized to lie between zero and one such that a higher number implies more political stability. Natural resource-exporting countries are those whose time series average of *NatRscWagric* is greater than or equal to 50% and those with less than 50% are classified as non-natural resource-exporting countries.

## 4. Empirical Methodology and Results

### 4.1. Fama-Macbeth-Type Regressions

**Hypothesis 1 (H1).** *Does our political risk-rating track net FDI flows i.e., are higher levels of political stability associated with higher net FDI inflows?*

Our baseline regression equation is as follows:

$$fdi_i = \alpha_0 + \alpha_1 Pol_{stab_i} + \alpha_2 Z_i + \varepsilon_i \tag{1}$$

where $fdi_i$ is the net FDI inflows as a percentage of GDP for country $i$, $Pol\_stab_i$ is the political stability score for country $i$ and $Z_i$ is a vector of control variables for country $i$.

Our control variables are the standard variables studied in the FDI literature. Consistent with existing literature, financial development is measured with the proxy variable domestic credit to private sector as a percentage of GDP; infrastructure of the host country is measured with the proxy the number of telephone lines per 100 people, albeit imperfectly, and gross fixed capital formation as a percentage of GDP. In addition, rising inflation is a proxy to measure macroeconomic policy inadequacies and the natural log of GDP per capita measures the size or attractiveness of the market. Finally, trade, which is the ratio of the sum of imports and exports to GDP, is a measure of the openness of an economy.

To test our hypotheses and to reduce the effect of cross-correlated residuals, we initially use Fama-Macbeth type regressions to estimate the coefficients and standard errors. In addition, FDI flows can fluctuate substantially year over year and this could lead to misleading results with regular panel data methods. Using Fama-Macbeth type regressions can alleviate this problem. Next, we also carry out dynamic panel data estimations.

First, we run cross-sectional regressions by year from 2000 to 2015. Then, we estimate and report the time-series averages for each coefficient. We estimate the standard errors by using the Newey-West standard error correction method to adjust for time-series autocorrelation. The results are found in Model 1 of Table 3 where the estimate of the parameter of interest, $\alpha_1$ is positive and statistically significant at the 1% level. To illustrate, let us consider two Sub-Saharan African countries in 2015 with different levels of political stability: Mauritius, the country with the highest political stability score (0.745) and Equatorial Guinea, the country with the lowest score (0.058). An improvement in political stability from the level of Equatorial Guinea to that of Mauritius will increase FDI by approximately 3.3%, i.e., $\frac{\partial FDI}{\partial PolStability} = (0.745 - 0.058) \times 4.74 \approx 3.26\%$.

**Hypothesis 2 (H2).** *Which components of Political Stability matter for FDI?*

We consider two specifications in Table 3: model [2] includes all the subcomponents and models [3] to [9] include one subcomponent at a time. In model [2], more rights and freedoms are associated with higher FDI flows whereas all of the other subcomponents either have the incorrect sign or are statistically insignificant. We suspect multicollinearity due to the generally high correlations between the subcomponents. We therefore include each subcomponent one at a time from models [3] to [9]. We find corruption, rights and freedoms, political constraints and governance quality to be the most important subcomponents of political risk that drive net FDI inflows. They are all statistically significant at the 1% level. To illustrate the economic significance:

- Let us compare Guinea Bissau, the country with the second worst corruption score in Sub-Saharan Africa to Botswana, the least corrupt in 2015 with normalized corruption scores of 0.016 and 0.679 respectively. An improvement in corruption from the level of Guinea Bissau to that of Botswana will increase FDI by approximately 2.49%, i.e., $\frac{\partial FDI}{\partial Corrupt} = (0.679 - 0.016) \times 3.761 \approx 2.49\%$. The above potential increase in net FDI flows is economically significant. For example, according to the 2017 World Investment Report published by UNCTAD, net FDI flows to Guinea Bissau increased by approximately 5.3% from 2015 to 2016. Another 2.5% increase in net FDI flows due to an improvement in their corruption score would have increased FDI flows to Guinea Bissau by almost 50%.

- Let us compare Zimbabwe, the second worst-performing country in Sub-Saharan Africa in terms of rights and freedoms to Ghana, the second best-performing country in 2015 with normalized rights and freedom scores of 0.112 and 0.782, respectively. An improvement in rights and freedoms from the level of Zimbabwe to that of Ghana will increase FDI by approximately 3.12%, i.e., $\frac{\partial FDI}{\partial RightsFreedom} = (0.782 - 0.112) \times 4.66 \approx 3.12\%$.

- Finally, let us compare Swaziland, the second worst performing country in Sub-Saharan Africa in terms of political constraints to South Africa, the third best-performing country in 2015 with normalized political constraint scores of 0.031 and 0.590, respectively. An improvement in political constraints from the level of Swaziland to that of South Africa will increase FDI by approximately 2%, i.e., $\frac{\partial FDI}{\partial PolConstraints} = (0.590 - 0.031) \times 3.604 \approx 2.01\%$.

Again, the economic significance cannot be overstated as countries such as Guinea Bissau, Zimbabwe and Swaziland generally struggle to maintain a positive year-on-year growth in net FDI inflows[8].

**Hypothesis 3 (H3).** *Do natural resources undermine the positive effect of a favorable political stability score on FDI?*

To answer this question, we interact our political stability variable with a natural resource exports variable to test the differential effects on FDI inflows for natural resource and non-natural resource-exporting countries. The natural resource exports variable is the

sum of fuel exports, agricultural raw materials, ores and metal exports as a percentage of merchandise exports in a given year.

**Table 3.** Political Risk and FDI.

| | | Dependent Variable = FDI | | | | | | | | |
|---|---|---|---|---|---|---|---|---|---|---|
| | | **[1]** | **[2]** | **[3]** | **[4]** | **[5]** | **[6]** | **[7]** | **[8]** | **[9]** |
| | Intercept | 5.141 | 1.885 | 5.764 | 5.629 | 3.834 | 4.342 | 4.751 | 2.663 | 5.419 |
| Financial Devt | DomCrdtoPriv (+) | 0.01 | 0.006 | 0.01 | 0.016 | 0.01 | 0.012 | 0.015 | 0.016 | 0.009 |
| Infrastructure | LogCellphones (+) | −1.502 | −1.449 * | −1.276 | −1.661 | −1.577 | −1.443 | −1.547 | −1.297 | −1.496 |
| | CapForm (+) | 0.135 ** | 0.140 ** | 0.132 ** | 0.128 ** | 0.139 ** | 0.135 ** | 0.127 ** | 0.132 ** | 0.128 ** |
| Trade | Trade (+) | 0.068 *** | 0.073 *** | 0.066 *** | 0.068 *** | 0.068 *** | 0.070 *** | 0.067 ** | 0.067 ** | 0.066 *** |
| Mkt size/attractives | LogGDP_PerCap10 (+) | −0.684 ** | −0.193 | −0.714 *** | −0.307 | −0.545 * | −0.568 ** | −0.376 * | −0.114 | −0.607 *** |
| Overall economic stability | Infl_deflator (−) | 0.042 ** | 0.046 *** | 0.034 *** | 0.021 * | 0.041 ** | 0.036 ** | 0.025 ** | 0.018 | 0.041 *** |
| Political Risk—Composite | PolStability | 4.740 *** | | | | | | | | |
| | Corruption | | 1.972 | 3.761 *** | | | | | | |
| | EthnoCultural | | −1.185 | | −0.854 | | | | | |
| Political Risk Components | RightsFreedom | | 10.877 *** | | | 4.655 *** | | | | |
| | PolConstraints | | −3.840 ** | | | | 3.604 *** | | | |
| | ConflictViol | | −6.511 *** | | | | | 1.171 | | |
| | WelfareSocioEco | | −0.536 | | | | | | −1.349 | |
| | GvceQuality | | 0.453 | | | | | | | 3.914 *** |
| N = # of years | | 16 | 16 | 16 | 16 | 16 | 16 | 16 | 16 | 16 |
| Total Sample Size | | 1984 | 1984 | 1984 | 1984 | 1984 | 1984 | 1984 | 1984 | 1984 |

This table presents Fama-McBeth regressions (with Newey-West adjusted standard errors) of FDI flows from 2000 to 2015. The dependent variable FDI, is net foreign direct investment as a percentage of GDP. DomCrdtoPriv is domestic credit to private sector as a percentage of GDP. LogCellphones is the natural log of mobile cellular subscriptions per 100 people. CapForm is gross fixed capital formation as a percentage of GDP. Trade is the sum of exports and import of goods and services as a percentage of GDP. LogGDP_PerCap10 is the natural log of GDP per capita in constant 2010 US dollars. Infl_deflator is the inflation, GDP deflator (annual %). The aforementioned variables are from the World Development Indicators published by the World Bank. PolStability is the political stability index proposed in this paper with the following seven components. Corruption is the corruption component, which measures the extent of corruption in a political system. EthnoCultural is the ethnocultural fractionalization component, which focuses on degree of ethnic, linguistic, religious and other cultural fractionalization. EthnoCulturalSq is the square of EthnoCultural. RightsFreedom is the rights and freedom component, which focuses on the availability and protection of human rights, political rights, and civil liberties. PolConstraints is the political constraints component, which focuses on the presence and effectiveness of checks and balances as well as the independence of the branches of government within a political system. ConflictViol is the conflict and violence component focuses on: (1) the existence of (2) the likelihood of (3) and the residual effect of past conflict and violence in a country (both internally and externally) and in surrounding countries. WelfareSocioEco is the welfare and socioeconomic conditions component, which is, based on the interaction of living standards, poverty levels, income disparities, and availability and access to social welfare programs. GvceQuality is the quality of political governance component, which reflects the people's perception of how confident they are in their government's capacity to provide quality public and civil services, the ability of the government to formulate and implement policies that can foster private sector growth, and confidence in their government guaranteeing the rule of law. PolStability and its seven components are all normalized to lie between zero and one such that a higher number implies more political stability. ***, ** and * denote statistical significance at 1%, 5% and 10%, respectively.

We estimate the following regression equation:

$$fdi_i = \alpha_0 + \alpha_1 Pol_{stab_i} + \alpha_2 nat + \alpha_3 nat \times Pol_{stab_i} + \alpha_4 Z_i + \varepsilon_i \tag{2}$$

Table 4 reports the results of the above equation. In model [1], the coefficient on both the political stability index and the natural resource variable are positive and significant at the 1% level, but the coefficient on our parameter of interest, the interaction between political stability and natural resource variable ($\alpha_3$) is negative and significant at the 5% level. This suggests that although the presence of natural resources may attract resource-seeking FDI, the presence of natural resources has a weakening effect on the positive relationship between FDI and political stability.

**Table 4.** Political Risk, Natural Resources, and FDI.

| | | Dependent Variable = FDI | | | | | | | | |
|---|---|---|---|---|---|---|---|---|---|---|
| | | **[1]** | **[2]** | **[3]** | **[4]** | **[5]** | **[6]** | **[7]** | **[8]** | **[9]** |
| | Intercept | 3.352 | 4.715 | 3.582 | 7.231 | 1.791 | 3.004 | 2.942 | 0.213 | 3.594 |
| Financial Devt | DomCrdtoPriv (+) | 0.011 | 0.008 | 0.007 | 0.014 | 0.012 | 0.012 | 0.011 | 0.008 | 0.004 |
| Infrastructure | LogCellphones (+) | −1.58 | −1.939 | −1.351 | −2.357 * | −1.63 | −1.676 | −1.972 | −1.717 * | −1.488 |
| | CapForm (+) | 0.097 | 0.090 * | 0.097 | 0.101 | 0.103 | 0.095 | 0.09 | 0.091 | 0.089 |
| Trade | Trade (+) | 0.068 *** | 0.071 *** | 0.064 *** | 0.067 *** | 0.068 *** | 0.069 *** | 0.065 *** | 0.065 *** | 0.063 *** |
| Mkt size/ attractives | LogGDP_PerCap10 (+) | −0.499 | −0.281 | −0.452 ** | 0.164 | −0.413 | −0.256 | 0.114 | 0.316 | −0.493 ** |
| Overall economic stability | Infl_deflator (−) | 0.026 | 0.061 | 0.021 | 0.023 | 0.027 | 0.018 | 0.011 | 0.008 | 0.036 |
| | NatRscWagric | 0.052 *** | −0.040 * | 0.055 *** | −0.058 *** | 0.056 *** | 0.037 *** | 0.037 *** | 0.042 *** | 0.057 *** |
| Political Risk— Composite | PolStability | 6.203 *** | | | | | | | | |
| | natAg_PolStab | −0.056 ** | | | | | | | | |
| | Corruption | | 11.183 * | 6.416 *** | | | | | | |
| | natAg_Corrupt | | −0.426 *** | −0.105 *** | | | | | | |
| | EthnoCultural | | −4.206 | | −5.898 ** | | | | | |
| | EthnoCulturalsq | | −0.835 | | 0.96 | | | | | |
| | natAg_EthnoCul | | 0.268 *** | | 0.241 *** | | | | | |
| | natAg_EthnoCulSq | | −0.178 ** | | −0.143 *** | | | | | |
| | RightsFreedom | | 20.687 * | | | 6.554 *** | | | | |
| | natAg_RghtFree | | −0.167 | | | −0.053 ** | | | | |
| Political Risk sub index | PolConstraints | | −11.358 | | | | 3.816 *** | | | |
| | natAg_PolCons | | 0.124 | | | | −0.029 | | | |
| | ConflictViol | | −10.984 * | | | | | 1.638 | | |
| | natAg_ConfViol | | 0.106 | | | | | −0.035 | | |
| | WelfareSocioEco | | −0.195 | | | | | | 1.246 | |
| | natAg_SocEcWelf | | 0.035 | | | | | | −0.048 *** | |
| | GvceQuality | | −2.032 | | | | | | | 7.484 *** |
| | natAg_GovQual | | 0.247 | | | | | | | −0.082 *** |
| N = # of years | | 16 | 16 | 16 | 16 | 16 | 16 | 16 | 16 | 16 |
| Total Sample Size | | 1984 | 1984 | 1984 | 1984 | 1984 | 1984 | 1984 | 1984 | 1984 |

This table presents Fama-McBeth regressions (with Newey-West adjusted standard errors) of FDI flows from 2000 to 2015. It assesses the direct effect of Political Risk and Natural Resources on FDI and assesses the interaction effect between Political Risk and Natural Resources on FDI. The dependent variable FDI is net foreign direct investment as a percentage of GDP. DomCrdtoPriv is domestic credit to private sector as a percentage of GDP. LogCellphones is the natural log of mobile cellular subscriptions per 100 people. CapForm is gross fixed capital formation as a percentage of GDP. Trade is the sum of exports and import of goods and services as a percentage of GDP. LogGDP_PerCap10 is the natural log of GDP per capita in constant 2010 US dollars. Infl_deflator is the inflation, GDP deflator (annual %). NatRscWagric is a natural resource variable, which is a country's sum of fuel exports, agricultural raw materials exports, and ores and metals as a percentage of merchandise exports. The aforementioned variables are from the World Development Indicators published by the World Bank. PolStability is the political stability index proposed in this paper with the following seven components. Corruption is the corruption component, which measures the extent of corruption in a political system. EthnoCultural is the ethno-cultural fractionalization component, which focuses on degree of ethnic, linguistic, religious and other cultural fractionalization. EthnoCulturalSq is the square of EthnoCultural. RightsFreedom is the rights and freedom component, which focuses on the availability and protection of human rights, political rights, and civil liberties. PolConstraints is the political constraints component, which focuses on the presence and effectiveness of checks and balances as well as the independence of the branches of government within a political system. ConflictViol is the conflict and violence component focuses on: (1) the existence of (2) the likelihood of (3) and the residual effect of past conflict and violence in a country (both internally and externally) and in surrounding countries. WelfareSocioEco is the welfare and socioeconomic conditions component, which is, based on the interaction of living standards, poverty levels, income disparities, and availability and access to social welfare programs. GvceQuality is the quality of political governance component, which reflects the people's perception of how confident they are in their government's capacity to provide quality public and civil services, the ability of the government to formulate and implement policies that can foster private sector growth, and confidence in their government guaranteeing the rule of law. PolStability and its seven components are all normalized to lie between zero and one such that a higher number implies more political stability. natAg_PolStab, natAg_Corrupt, natAg_EthnoCul, natAg_EthnoCulSq, RightsFreedom, natAg_RghtFree, natAg_PolCons, natAg_ConfViol, natAg_SocEcWelf, and natAg_GovQual are interaction terms between NatRscWagric and PolStability, Corruption, EthnoCultural, EthnoCulturalsq, RightsFreedom, PolConstraints, ConflictViol, WelfareSocioEco, and GvceQuality respectively. ***, ** and * denote statistical significance at 1%, 5% and 10% respectively.

The above equation is re-estimated for each of our political stability subcomponents as well, i.e., by interacting our natural resource variable with the various subcomponents. We do find that natural resources do indeed reduce the positive effect of a politically stable environment on net FDI inflows.

Models [3] through [9] look at which components of political stability are undermined by the presence of natural resources. In model [3], the coefficient on the corruption component is positive and significant at the 1% level and the coefficient on the interaction between corruption and natural resources is negative and significant at the 1% level. This suggests that the presence of natural resources makes less corruption less effective in attracting FDI flows. The same holds with the rights and freedoms, and the quality of political governance components in models [5] and [9] respectively. The interaction between natural resources and the socioeconomic welfare variable is negative and significant at the 1% level despite the absence of a significant relationship between the socioeconomic welfare variable and FDI.

In model [4], we included a quadratic term for ethnocultural fractionalization to account for a possible nonlinear effect and interacted our natural resource variable with both the linear and quadratic term. As posited in the introduction, the justification stems from the conflicting arguments in the political economy literature that suggests that both high and low levels of ethnocultural fractionalization could be detrimental to FDI.

The coefficient on ethnocultural is negative and significant whereas the coefficient on the quadratic term is insignificant. However, the interaction of the natural resource variable and ethnocultural fractionalization is positive and significant at the 1% level, and the interaction of the natural resource variable and the quadratic ethnocultural term is negative and significant at the 1% level[9]. This may suggest that for natural resource-exporting countries, there is some optimal level of ethnocultural fractionalization that may be conducive for attracting FDI.

### 4.2. Dynamic Panel Data Regressions

Busse and Hefeker (2007) point out a few problems with standard FDI estimations. The first is endogeneity of the regressors; for instance, trade and gross fixed capital formation may well be influenced by FDI. Second, because FDI flows may fluctuate from year to year, reporting time series averages of parameter estimates may not properly account for significant deviations of variables from the mean for that period. Panel data methods that allow for some variation in the time series dimension of the data whilst at the same time controlling for potential wild fluctuations in FDI flows, endogeneity of regressors and finally also controlling for the autocorrelation of the disturbances is employed in our estimations here. We employ dynamic panel data estimation because as Busse and Hefeker (2007) point out, a regular fixed effects model will not address the issue of the endogeneity of the regressors.

To reduce the effect of wild fluctuations in year-on-year FDI flows (and of other variables), we do not use annual observations for all variables but 3-year averages, which significantly reduces our sample size. This is a common practice in the FDI literature. To mitigate the effect of the autocorrelation of residuals because of the time series component, we include the lag of FDI as an independent variable. In addition to solving an econometric problem, it can be argued that the decision by multinationals to invest or not in a particular country is also influenced by the previous years' FDI to that country. As a result, several FDI studies include the lag of FDI as an explanatory variable, e.g., Asiedu and Lien (2011). This results in a dynamic panel model.

We use the difference GMM estimator proposed by Arellano and Bond (1991) and the system GMM estimator proposed by Blundell and Bond (1998) to estimate our dynamic panel data model. The independent variables are treated as exogenous in all of our models and we do not use external instruments. For both models we use the two-step GMM estimator which is robust to heteroscedasticity and asymptotically efficient. We also test for second order serial correlation since both estimators require no second order serial

correlation in the residuals. For both models, Sargan tests of over-identifying restrictions are performed to test for the overall validity of our instruments.

We revisit the relationship between political stability and FDI using dynamic panel data models in Table 5 below and to conserve space we only report the parameter estimates of the political risk variables[10].

The results from Table 5 under both the difference GMM Estimator and the system GMM Estimator align with our earlier findings, i.e., that our political stability index tracks FDI flows and that there exists a nonlinear concave relationship between net FDI inflows and the degree of ethnocultural fractionalization. Under the difference GMM estimator (Arellano and Bond (1991)), corruption, ethnocultural fractionalization, rights and freedom, socioeconomic welfare and governance quality are all significant at the 1% level. Under the system GMM estimator (Blundell and Bond (1998)), all of our political stability subcomponents are significant (at the 1% level), with the exception of socioeconomic welfare. Under both estimation models, we see a nonlinear concave relationship between net FDI flows and ethnocultural fractionalization.

Next, we revisit the question of whether natural resources moderate the relationship between FDI and political risk using dynamic panel data models. Results are reported in Table 6 below and to conserve space we only report the parameter estimates of the political risk variables[11]. Under the difference GMM, we see that natural resources moderate the relationship between FDI and the following: corruption, ethnocultural fractionalization, rights and freedoms, socioeconomic welfare and governance quality. Under the system GMM, the case is true for all our political stability subcomponents. That is, the presence of natural resources weakens the positive relationship between FDI and all our political stability subcomponents. Overall, our main results that the presence of natural resources moderates the relationship between FDI and political risk holds. Authors such as Acemoglu et al. (2005); Bobba and Coviello (2007) have shown that the difference GMM and system GMM can produce different results. That notwithstanding, our results under both estimation methods support our initial findings.

It is also worth mentioning that there is no evidence of second order serial correlation. In addition, although not reported in the tables, the Sargan tests of over-identifying restrictions (the null hypothesis that the instrument set is appropriate for the data at hand) all have *p*-values less than 1, thus rejecting the null. Typically, this would indicate that there is a problem with the instrument set. However, Blundell et al. (2003, p. 86) found that the Sargan test tends to over reject in small samples; and Bowsher (2002) documents the poor quality of the Sargan tests " . . . in panels of dimensions that are commonly encountered in empirical work." Finally, Sargan tests are known to over reject when you have significant individual effects, which are evident in our data. Thus, the small *p*-values are probably not cause for much concern.

### *4.3. Informational Content of the Political Stability Index*

A natural question to ask is whether our political stability index (referred to as PolStability) is related to well-known indices, such as the ICRG's political risk index (referred to as *PR_ICRG*), and more importantly, if the political stability index contains any unique information which can be used to explain FDI flows.

**Table 5.** Political Risk and FDI Revisited—Dynamic Panel Data Models.

| | | Panel a: Difference GMM | | | | | | | | |
|---|---|---|---|---|---|---|---|---|---|---|
| | | [1] | [2] | [3] | [4] | [5] | [6] | [7] | [8] | [9] |
| Political Risk—Composite | PolStability | 11.853 *** | | | | | | | | |
| Political Risk Components | Corruption | | 0.916 *** | 6.244 *** | | | | | | |
| | EthnoCultural | | 26.317 *** | | 14.100 *** | | | | | |
| | EthnoCulturalsq | | −18.639 *** | | −9.723 *** | | | | | |
| | RightsFreedom | | 14.512 *** | | | 11.232 *** | | | | |
| | PolConstraints | | −7.713 *** | | | | 0.272 | | | |
| | ConflictViol | | −3.208 *** | | | | | 0.828 | | |
| | WelfareSocioEco | | 4.213 *** | | | | | | 7.939 *** | |
| | GvceQuality | | 21.012 *** | | | | | | | 20.901 *** |
| Serial correlation test (*p*-value) | | 0.210 | 0.202 | 0.226 | 0.212 | 0.216 | 0.216 | 0.212 | 0.204 | 0.201 |
| Number of observations | | 680 | 680 | 680 | 680 | 680 | 680 | 680 | 680 | 680 |
| Number of countries, n | | 136 | 136 | 136 | 136 | 136 | 136 | 136 | 136 | 136 |
| | | Panel b: System GMM | | | | | | | | |
| | | [1] | [2] | [3] | [4] | [5] | [6] | [7] | [8] | [9] |
| Political Risk—Composite | PolStability | 3.789 *** | | | | | | | | |
| Political Risk Components | Corruption | | 1.598 *** | 2.280 *** | | | | | | |
| | EthnoCultural | | −0.775 *** | | 0.360 *** | | | | | |
| | EthnoCulturalsq | | −1.547 *** | | −1.885 *** | | | | | |
| | RightsFreedom | | 9.393 *** | | | 4.267 *** | | | | |
| | PolConstraints | | −3.312 *** | | | | 2.982 *** | | | |
| | ConflictViol | | −6.103 *** | | | | | 0.867 *** | | |
| | WelfareSocioEco | | −1.306 *** | | | | | | 0.667 | |
| | GvceQuality | | 0.741 *** | | | | | | | 3.278 *** |
| Serial correlation test (*p*-value) | | 0.231 | 0.233 | 0.233 | 0.228 | 0.232 | 0.232 | 0.230 | 0.228 | 0.230 |
| Number of observations | | 680 | 680 | 680 | 680 | 680 | 680 | 680 | 680 | 680 |
| Number of countries, n | | 136 | 136 | 136 | 136 | 136 | 136 | 136 | 136 | 136 |

This table presents difference GMM estimator (Arellano and Bond 1991) and system GMM estimator (Blundell and Bond 1998) results in panel a and panel b, respectively. The dependent variable FDI, is net foreign direct investment as a percentage of GDP. The following control variables are used although their parameter estimates are not reported: lag of the dependent variable, FDI; domestic credit to private sector as a percentage of GDP; the natural log of Mobile cellular subscriptions per 100 people; gross fixed capital formation as a percentage of GDP; trade defined as the sum of exports and import of goods and services as a percentage of GDP; the natural log of GDP per capita in constant 2010 US dollars; inflation i.e., the GDP deflator (annual %). The aforementioned variables are from the World Development Indicators published by the World Bank. *PolStability* is the political stability index proposed in this paper with the following seven components. *Corruption* is the corruption component which measures the extent of corruption in a political system. *EthnoCultural* is the ethno-cultural fractionalization component which focuses on degree of ethnic, linguistic, religious and other cultural fractionalization. *EthnoCulturalSq* is the square of *EthnoCultural*. *RightsFreedom* is the rights and freedom component which focuses on the availability and protection of human rights, political rights, and civil liberties. *PolConstraints* is the political constraints component which focuses on the presence and effectiveness of checks and balances as well as the independence of the branches of government within a political system. *ConflictViol* is the conflict and violence component focuses on: (1) the existence of (2) the likelihood of (3) and the residual effect of past conflict and violence in a country (both internally and externally) and in surrounding countries. *WelfareSocioEco* is the welfare and socioeconomic conditions component which is based on the interaction of living standards, poverty levels, income disparities, and availability and access to social welfare programs. *GvceQuality* is the quality of political governance component which reflects the people's perception of how confident they are in their government's capacity to provide quality public and civil services, the ability of the government to formulate and implement policies that can foster private sector growth, and confidence in their government guaranteeing the rule of law. *PolStability* and its seven components are all normalized to lie between zero and one such that a higher number implies more political stability. ***, ** and * denote statistical significance at 1%, 5% and 10% respectively.

**Table 6.** Political Risk, Natural Resources, and FDI Revisited—Dynamic Panel Data Models.

| | | **Panel a: Difference GMM** | | | | | | | | |
| | | **[1]** | **[2]** | **[3]** | **[4]** | **[5]** | **[6]** | **[7]** | **[8]** | **[9]** |
| | NatRscWagric | 0.003 *** | 0.006 *** | 0.004 *** | 0.002 *** | 0.002 *** | 0.001 *** | 0.003 *** | 0.002 *** | 0.004 *** |
| Political Risk—Composite | PolStability | 7.543 *** | | | | | | | | |
| | natAg_PolStab | −0.005 *** | | | | | | | | |
| | Corruption | | −9.314 *** | 0.460 *** | | | | | | |
| | natAg_Corrupt | | 0.008 *** | −0.008 *** | | | | | | |
| | EthnoCultural | | 21.999 *** | | 7.473 *** | | | | | |
| | EthnoCulturalsq | | −14.089 *** | | 0.519 * | | | | | |
| | natAg_EthnoCul | | 0.009 *** | | 0.013 *** | | | | | |
| | natAg_EthnoCulSq | | −0.007 *** | | −0.014 *** | | | | | |
| | RightsFreedom | | 10.875 *** | | | 7.840 *** | | | | |
| Political Risk sub index | natAg_RghtFree | | 0.002 *** | | | −0.004 *** | | | | |
| | PolConstraints | | −0.799 *** | | | | −1.153 *** | | | |
| | natAg_PolCons | | −0.002 *** | | | | 0.002 | | | |
| | ConflictViol | | −1.882 *** | | | | | −6.106 * | | |
| | natAg_ConfViol | | 0.000 | | | | | 0.005 ** | | |
| | WelfareSocioEco | | 11.725 *** | | | | | | 4.206 *** | |
| | natAg_SocEcWelf | | −0.004 *** | | | | | | −0.003 *** | |
| | GvceQuality | | 16.157 *** | | | | | | | 11.903 *** |
| | natAg_GovQual | | 0.003 *** | | | | | | | −0.007 *** |
| Serial correlation test (*p*-value) | | 0.213 | 0.211 | 0.231 | 0.224 | 0.218 | 0.220 | 0.216 | 0.202 | 0.196 |
| Number of observations | | 665 | 665 | 665 | 665 | 665 | 665 | 665 | 665 | 665 |
| Number of countries, n | | 133 | 133 | 133 | 133 | 133 | 133 | 133 | 133 | 133 |
| | | **Panel b: System GMM** | | | | | | | | |
| | | **[1]** | **[2]** | **[3]** | **[4]** | **[5]** | **[6]** | **[7]** | **[8]** | **[9]** |
| | NatRscWagric | 0.001 *** | 0.001 *** | 0.001 *** | 0.001 *** | 0.001 | 0.001 *** | 0.001 *** | 0.001 *** | 0.001 *** |
| Political Risk—Composite | PolStability | 4.693 *** | | | | | | | | |
| | natAg_PolStab | −0.002 *** | | | | | | | | |
| | Corruption | | 5.177 *** | 3.657 *** | | | | | | |
| | natAg_Corrupt | | −0.003 *** | −0.003 *** | | | | | | |
| | EthnoCultural | | −2.192 *** | | 2.233 *** | | | | | |
| | EthnoCulturalsq | | 0.359 ** | | −3.526 *** | | | | | |
| | natAg_EthnoCul | | 0.002 *** | | 0.001 *** | | | | | |
| | natAg_EthnoCulSq | | −0.001 *** | | −0.001 *** | | | | | |
| | RightsFreedom | | 12.683 *** | | | 4.109 *** | | | | |
| Political Risk sub index | natAg_RghtFree | | −0.002 *** | | | −0.001 *** | | | | |
| | PolConstraints | | −6.104 *** | | | | 3.607 *** | | | |
| | natAg_PolCons | | 0.002 *** | | | | −0.001 *** | | | |
| | ConflictViol | | −8.695 *** | | | | | 1.414 *** | | |
| | natAg_ConfViol | | 0.002 *** | | | | | −0.001 *** | | |
| | WelfareSocioEco | | 6.617 *** | | | | | | 2.599 *** | |
| | natAg_SocEcWelf | | −0.004 *** | | | | | | −0.002 *** | |
| | GvceQuality | | −5.895 *** | | | | | | | 4.004 *** |
| | natAg_GovQual | | 0.004 *** | | | | | | | −0.006 *** |
| Serial correlation test (*p*-value) | | 0.225 | 0.227 | 0.229 | 0.223 | 0.226 | 0.224 | 0.224 | 0.222 | 0.223 |
| Number of observations | | 665 | 665 | 665 | 665 | 665 | 665 | 665 | 665 | 665 |
| Number of countries, n | | 133 | 133 | 133 | 133 | 133 | 133 | 133 | 133 | 133 |

This table presents difference GMM estimator (Arellano and Bond 1991) and system GMM estimator (Blundell and Bond 1998) results in panel a and panel b respectively. The dependent variable FDI, is net foreign direct investment as a percentage of GDP. The following control variables are used although their parameter estimates are not reported: lag of the dependent variable, FDI; domestic credit to private sector as a percentage of GDP; the natural log of Mobile cellular subscriptions per 100 people; gross fixed capital formation as a percentage of GDP; trade defined as the sum of exports and import of goods and services as a percentage of GDP; the natural log of GDP per capita in

constant 2010 US dollars; inflation i.e., the GDP deflator (annual %). The aforementioned variables are from the World Development Indicators published by the World Bank. PolStability is the political stability index proposed in this paper with the following seven components. Corruption is the corruption component, which measures the extent of corruption in a political system. EthnoCultural is the ethno-cultural fractionalization component, which focuses on degree of ethnic, linguistic, religious and other cultural fractionalization. EthnoCulturalSq is the square of EthnoCultural. RightsFreedom is the rights and freedom component, which focuses on the availability and protection of human rights, political rights, and civil liberties. PolConstraints is the political constraints component, which focuses on the presence and effectiveness of checks and balances as well as the independence of the branches of government within a political system. ConflictViol is the conflict and violence component focuses on: (1) the existence of (2) the likelihood of (3) and the residual effect of past conflict and violence in a country (both internally and externally) and in surrounding countries. WelfareSocioEco is the welfare and socioeconomic conditions component, which is, based on the interaction of living standards, poverty levels, income disparities, and availability and access to social welfare programs. GvceQuality is the quality of political governance component, which reflects the people's perception of how confident they are in their government's capacity to provide quality public and civil services, the ability of the government to formulate and implement policies that can foster private sector growth, and confidence in their government guaranteeing the rule of law. PolStability and its seven components are all normalized to lie between zero and one such that a higher number implies more political stability. natAg_PolStab, natAg_Corrupt, natAg_EthnoCul, natAg_EthnoCulSq, RightsFreedom, natAg_RghtFree, natAg_PolCons, natAg_ConfViol, natAg_SocEcWelf, and natAg_GovQual are interaction terms between NatRscWagric and PolStability, Corruption, EthnoCultural, EthnoCulturalsq, RightsFreedom, PolConstraints, ConflictViol, WelfareSocioEco, and GvceQuality respectively. ***, ** and * denote statistical significance at 1%, 5% and 10% respectively.

We begin by running pooled OLS with *PolStability* as the dependent variable and *PR_ICRG* as the independent variable. We obtain an R-Square of 0.646 and a correlation coefficient between both variables of 0.804. This implies approximately 65% of the variation in the political stability index proposed in this study is explained by the variability in the ICRG's political risk rating (OLS with year and country effects generates an R-Square of 0.984). Thus, both measures seem to have some common information. To determine if the political stability index picks up any unique information, not contained in the ICRG political risk rating, which can explain FDI flows, we carry out the following tests (the results are documented in Table 7):

- Models [1] and [2] include *PolStability* and *PR_ICRG*, respectively separately while model [3] includes both of them. We find that including the measures separately, the coefficient on both are statistically significant at the 1% level. When we include both measures in the same regressions, the coefficient on the political stability index is positive and statistically significant at the 1% level while that of *PR_ICRG* is positive and statistically significant at the 5% level. While the results so far are encouraging for our political stability index, we carry out further tests to determine whether it contains unique information.
- In Model [4] we start by estimating OLS regressions with *PolStability* as the dependent variable and *PR_ICRG* as the independent variable by year from 2000 to 2015 and we save the predicted values and the residual (both are orthogonal). Next, we include the predicted values and residuals in model [4]. The predicted values should have political risk information that is common to both risk measures while the residuals should have political risk information that is unique to the political stability index proposed in this study. The coefficient on the residual is positive (3.478) and significant at the 1% level and it is the same coefficient on the *PolStability* variable in model [3]. The coefficient of the predicted values is also positive and statistically significant. This would suggest that although the common information set between this paper's political stability index and the ICRG's political risk rating does explain net FDI inflows, the unique and incremental information contained in the political stability index has the ability to explain net FDI flows beyond that explained by the ICRG's political risk index.
- Similarly in Model [5], we start by estimating OLS regressions with *PR_ICRG* as the dependent variable and *PolStability* as the independent variable by year from 2000 to 2015 and we save the predicted values and the residual (both are orthogonal). Next, we include the predicted values and residuals in model [5]. The predicted values should have political risk information that is common to both risk measures while the

residuals should have political risk information that is unique to *PR_ICRG*. This time the coefficient on the predicted value is positive and significant at the 1% level while the coefficient on the residual is positive and significant at the 5% level.

The coefficient on the residuals in models [4] and [5] are the same coefficients on the *PolStability* and *PR_ICRG* variables in model [3], suggesting that while both measures are correlated and share common information, each measure has unique information that it contributes to explain FDI inflows. However, the political stability index's unique information is more statistically significant (at the 1% level while the ICRG's is at the 5% level), suggesting it may have more reliable unique information and thus is a political risk proxy in its own right.

**Table 7.** Information Content of the Political Stability vis-à-vis ICRG's Political Risk Score.

| Hypotheses | Variables | Dependent Variable = FDI | | | | |
|---|---|---|---|---|---|---|
| | | **[1]** | **[2]** | **[3]** | **[4]** | **[5]** |
| | Intercept | 8.290 | 5.566 | 6.62 | 9.547 | 2.165 |
| Financial Devt | DomCrdtoPriv | 0.009 | 0.009 | 0.008 | 0.008 | 0.008 |
| Infrastructure | logCellphones | −2.214 * | −2.484 * | −2.302 * | −2.302 * | −2.302 * |
| | CapForm | 0.101 * | 0.089 | 0.095 | 0.095 | 0.095 |
| Trade | trade | 0.072 *** | 0.067 *** | 0.069 *** | 0.069 *** | 0.069 *** |
| Mkt size/ attractives | logGDP_PerCap10 | −0.633 * | −0.723 * | −0.853 ** | −0.853 ** | −0.853 ** |
| Overall economic stability | Infl_deflator | 0.0470 ** | 0.036 ** | 0.051 ** | 0.051 ** | 0.051 ** |
| Political Risk | PolStability | 5.188 *** | | 3.478 *** | | |
| | PR_ICRG | | 11.538 *** | 7.376 ** | | |
| | pred: y = polStab, X = ICRG | | | | 8.010 *** | |
| | resid: y = polStab, X = ICRG | | | | 3.478 *** | |
| | pred: y = ICRG, X = PolStability | | | | | 16.075 *** |
| | resid: y = ICRG, X = PolStability | | | | | 7.376 ** |
| N = # Years | | 16 | 16 | 16 | 16 | 16 |
| # Countries | | 121 | 121 | 121 | 121 | 121 |

This table presents Fama-McBeth regressions of FDI flows from 2000 to 2015. The dependent variable FDI, is net foreign direct investment as a percentage of GDP. DomCrdtoPriv is domestic credit to private sector as a percentage of GDP. LogCellphones is the natural log of mobile cellular subscriptions per 100 people. CapForm is gross fixed capital formation as a percentage of GDP. Trade is the sum of exports and import of goods and services as a percentage of GDP. LogGDP_PerCap10 is the natural log of GDP per capita in constant 2010 US dollars. Infl_deflator is the inflation, GDP deflator (annual %). The aforementioned variables are from the World Development Indicators published by the World Bank. PolStability is the political stability index proposed in this paper and it is normalized to lie between zero and one such that a higher number implies more political stability. PR_ICRG is the political risk rating, which is the sum of the annual averages of the 12 political risk-rating components (Government Stability, Socioeconomic Conditions, Investment Profile, Internal Conflict, External Conflict, Corruption, Military in Politics, Religious Tensions, Law and Order, Ethnic Tensions, Democratic Accountability, Bureaucracy Quality) from ICRG's researcher's dataset. This variable is also normalized to lie between zero and one such that a higher number implies less political risk. Pred: Y = PolStability, X = PR_ICRG are the predicted values of annual regressions where PolStability is the dependent variable and PR_ICRG is the independent variable. resid: Y = PolStability, X = PR_ICRG are the residual of annual regressions where PolStability is the dependent variable and PR_ICRG is the independent variable. resid: Y = PR_ICRG, X = PolStability are the residual values of annual regressions where PR_ICRG is the dependent variable and PolStability is the independent variable. pred: Y = PR_ICRG, X = PolStability are the predicted values of annual regressions where PR_ICRG is the dependent variable and PolStability is the independent variable. ***, ** and * denote statistical significance at 1%, 5% and 10% respectively.

## 5. Conclusions

This article examines the relationship between a novel measure of political risk and FDI. We show that this novel measure contains unique and incremental information that is able to explain net FDI flows, beyond that studied in the literature. We find in this study that the presence of natural resources can moderate the positive relationship between FDI and our political stability index. That is, in the presence of natural resources, political stability may not matter as much to multinationals as it would be in the absence of natural

resources. This aligns with the conclusions of Asiedu and Lien (2011) who show that the presence of natural resources moderates the relationship between FDI and democracy. Interestingly, we also find some evidence that a concave relationship exists between net FDI inflows and the degree of ethnocultural fractionalization, especially for natural resource-exporting countries. This may imply that beyond some optimal level of ethnocultural fractionalization, the attractiveness of natural resource-rich countries may diminish for foreign direct investment capital.

This study contributes to the political economy literature by introducing a new index, the political stability index, which draws inspiration from Karolyi (2015). This index attempts to summarize or capture the underlying fragilities of a country's political system, which could affect investors. While this index is not the only index out there (for example the ICRG's political risk index), we argue that it is a robust index and contains relevant, unique and incremental information. We show that this unique information explains net FDI flows.

We also break down the political stability index into seven subcomponents[12] to disentangle the sources of political risk. For example, several countries may have similar political stability scores, but the underlying risk factors may be different. Thus, what may be important in certain instances is not the political stability score of a country but the source of underlying fragilities in a political system. This adds some versatility to the political risk measures used in this paper.

The ICRG's political risk index has been widely used as a proxy for political risk in FDI studies. We contend that the political stability index used in this paper is different from the ICRG's for the following reasons. The aim of the ICRG's political risk measure is to determine the political stability of a country by generating risk ratings based on a subjective analysis of political information. On the other hand, the goal of the political stability index used in this paper is to assess the fundamental fragilities of a country's political system based on a wide variety of variables established in the political economy literature that purport to measure various aspects of political risk. Thus, instead of relying on the subjective assessment of various country experts, this index relies on a host of variables provided by a range of entities to come up with a robust political risk assessment. This index covers 150 countries while the ICRG covers 140. Both indexes have 130 countries in common, while the ICRG has 10[13] countries that this index does not track, and this index has 20[14] countries that the ICRG does not track.

The next step in this research is to extend tests of the political stability index to examine whether it can explain market phenomena like stock market and currency market returns. We believe this freely-available index should be of interest to policymakers seeking to attract more foreign direct investments and foreign portfolio investments, as well as investors seeking to invest in foreign markets, especially emerging markets.

**Author Contributions:** Conceptualization, Pavel Jeutang and Kwabena Kesse; methodology, Pavel Jeutang and Kwabena Kesse; formal analysis, Pavel Jeutang; data curation, Pavel Jeutang; writing—original draft preparation, Kwabena Kesse; writing—review and editing, Pavel Jeutang and Kwabena Kesse. All authors have read and agreed to the published version of the manuscript.

**Funding:** This research received no external funding.

**Institutional Review Board Statement:** Not applicable.

**Informed Consent Statement:** Not applicable.

**Data Availability Statement:** Our political risk estimates can be accessed at https://www.dropbox.com/sh/hwo3rucpf5o7y50/AADTQCleBhUKizalZz8v_k0fa?dl=0, accessed date 4 June 2021. Other variables are extracted from the World Bank's World Development Indicators.

**Conflicts of Interest:** The authors declare no conflict of interest.

## Appendix A

We provide a database of the political risk estimates at the link below:

Database of Political Risk estimates—150 countries from 2000 to 2015. The data was accessed on January 12, 2017

https://www.dropbox.com/sh/hwo3rucpf5o7y50/AADTQCleBhUKizalZz8v_k0fa?dl=0 (accessed on 1 October 2021)

We also provide a brief description of the index below:

*The Political Stability Index*

The political stability index is a political risk measure that seeks to quantify or provide a numerical measure of the underlying fragilities of country's political system. This measure generates a robust political risk assessment of a country by relying on a wide variety of variables (both subjective and objective) established in the political economy and political science literature that purport to measure various aspects of political risk. This index and its subcomponents could be of interest to academics, investors, and policy makers. The political stability index is attractive for the following reasons:

The underlying variables that go into the construction of this index are mostly from the existing academic literature on political economy and usually measure some underlying aspect of political risk. However, some of the underlying variables are also from well-known agencies and have been used extensively in academic literature. This political stability index draws from 18 sources pulling from 21 databases and uses 50 underlying variables. Here is the comprehensive list of data sources:

1. Center for Systemic Peace
2. Corruption Perceptions Index—Transparency International
3. Database of Political Institutions—World Bank (Beck et al. 2001; Keefer and Stasavage 2003; Pagano and Volpin 2005)
4. Ethnic and Cultural Diversity by Country (Fearon 2003)
5. Fractionalization (Alesina et al. 2003)
6. Fragile States Index—Fund for Peace
7. Freedom of the World—Freedom House
8. Global Terrorism Index—Vision of Humanity
9. Human Development Index—United Nations Development Program
10. Index of Economic Freedom—Heritage Foundation
11. Press Freedom—Reporters Sans Frontières
12. Political Terror Scale Dataset (Gibney 2002)
13. The Cingranelli-Richards (CIRI) human rights dataset (Cingranelli and Richards 2010)
14. The Quality of Government Institute (Dahlberg et al. 2017)
15. The Political Constraint Index (POLCON) Dataset (Henisz 2000)
16. The Worldwide Governance Indicators (WGI) project
17. Varieties of Democracy (V-Dem) Project (Coppedge et al. 2016)
18. World Development Indicators (WDI)—World Bank

The political stability index can be broken down into the following seven subcomponents that capture different aspects of political risk:

1. **Conflict and Violence:** The Conflict and Violence Subcomponent focuses on: (1) the existence of, (2) the likelihood of, and (3) the residual effect of past conflict and violence in a country (both internally and externally) and in surrounding countries. It includes the following:

   - The perception of the likelihood of politically-motivated violence, including terrorism
   - The fractionalization of the country's territory
   - The state's capacity to manage conflict
   - The incidence of terrorism
   - The country's involvement in international war and violence
   - Civil violence, civil war, ethnic violence and ethnic war involving the state
   - Conflict and violence in bordering states

- Political terror including political imprisonment, political torture and political disappearance.

2.  **Corruption:** The Corruption subcomponent measures the extent of corruption in a political system. It is based on the perceived level of corruption and the prevalence of political corruption within the executive, legislative and judiciary branches. It captures different facets of corruption including bribery, embezzlement, exercise of power for private gain and the influence of the law making process and the exercise of law.

3.  **Ethnocultural Fragilities:** The Ethnocultural Fragilities subcomponent focuses on ethnic, linguistic, religious and other cultural fractionalization. It is based on cultural distances across different ethnic, linguistic and religious groups as well as the likelihood of people being from different ethnic, linguistic and religious groups within a country.

4.  **Political Constraints:** The Political Constraints subcomponent focuses on the presence and effectiveness of checks and balances as well as the independence of the branches of government within a political system. It is based on constraints on the executive, entrenchment of the executive, control or influence of the legislature by the ruling party, the effectiveness of the opposition in the legislature, independence of the judiciary, electoral competitiveness and the overall quality and extent of democracy.

5.  **Quality of Political Governance:** The Quality of Political Governance subcomponent reflects the people's perception of how confident they are in their government's capacity to provide quality public and civil services, the ability of the government to formulate and implement policies that can foster private sector growth and confidence in their government guaranteeing the rule of law.

6.  **Rights and Freedom:** The Rights and Freedom subcomponent focuses on the availability and protection of human rights, political rights and civil liberties. It includes:

    - Freedom of speech and of expression
    - Freedom of belief and religion
    - Freedom of the press
    - Freedom of academic and cultural expression
    - The rule of law and access to justice
    - Freedom from torture and executions
    - Freedom of assembly and association (in political parties, trade unions, cultural organizations, or other special-interest groups and organizations)
    - The ability to participate freely in the political process (including the right to vote freely in legitimate elections, compete for public office, and elect representatives who are accountable to the electorate)
    - Personal autonomy without interference from the state, freedom of domestic and foreign movement, and freedom from political killings.

7.  **Welfare and Socioeconomic Conditions:** The Welfare and Socioeconomic Conditions subcomponent is based on the interaction of living standards, poverty levels, income disparities and availability and access to social welfare programs.

The political stability index is a robust measure of political risk since every subcomponent relies on a variety of underlying variables, from a variety of data sources and databases, which in turn employ a variety of estimation techniques (both objective and subjective). For example, the political constraints subcomponent is based on sixteen variables from seven different data sources. More information on how the political stability index is developed is available upon request from the authors.

The version of the political stability index used in this study covers 150 countries from 2000 to 2017 and includes following countries grouped by regions[15]:

**Table A1.** Western Europe and North America (including Cyprus, Australia and New Zealand)—22 Countries.

| | | | |
|---|---|---|---|
| Australia | Finland | Luxembourg | Sweden |
| Austria | France | Netherlands | Switzerland |
| Belgium | Germany | New Zealand | United Kingdom |
| Canada | Greece | Norway | United States |
| Cyprus | Ireland | Portugal | |
| Denmark | Italy | Spain | |

**Table A2.** Latin America (including Cuba and the Dominican Republic)—19 Countries.

| | | | |
|---|---|---|---|
| Argentina | Costa Rica | Guatemala | Paraguay |
| Bolivia | Cuba | Honduras | Peru |
| Brazil | Dominican Republic | Mexico | Uruguay |
| Chile | Ecuador | Nicaragua | Venezuela |
| Colombia | El Salvador | Panama | |

**Table A3.** Eastern Europe and Central Asia (post-Communist; including Mongolia)—26 Countries.

| | | | |
|---|---|---|---|
| Albania | Estonia | Macedonia | Slovenia |
| Armenia | Georgia | Moldova | Tajikistan |
| Azerbaijan | Hungary | Mongolia | Turkmenistan |
| Belarus | Kazakhstan | Poland | Ukraine |
| Bulgaria | Kyrgyzstan | Romania | Uzbekistan |
| Croatia | Latvia | Russia | |
| Czech Republic | Lithuania | Slovakia | |

**Table A4.** East Asia, South-East Asia, South Asia and The Pacific (excluding Australia and New Zealand)—18 Countries.

| | | | |
|---|---|---|---|
| China | Laos | Thailand | Pakistan |
| Japan | Malaysia | Vietnam | Sri Lanka |
| South Korea | Myanmar | Bangladesh | Papua New Guinea |
| Cambodia | Philippines | India | |
| Indonesia | Singapore | Nepal | |

**Table A5.** The Middle East and North Africa (including Israel and Turkey)—18 Countries.

| | | | |
|---|---|---|---|
| Algeria | Jordan | Oman | Turkey |
| Bahrain | Kuwait | Qatar | UAE |
| Egypt | Lebanon | Saudi Arabia | Yemen |
| Iran | Libya | Syria | |
| Israel | Morocco | Tunisia | |

**Table A6.** Sub-Saharan Africa—42 Countries.

| | | | |
|---|---|---|---|
| Angola | Djibouti | Madagascar | Sierra Leone |
| Benin | Equatorial Guinea | Malawi | South Africa |
| Botswana | Ethiopia | Mali | Sudan |
| Burkina Faso | Gabon | Mauritania | Swaziland |
| Burundi | Gambia | Mauritius | Tanzania |
| Cameroon | Ghana | Mozambique | Togo |
| Central African Republic | Guinea | Namibia | Uganda |
| Chad | Guinea-Bissau | Niger | Zambia |
| Congo Brazzaville | Kenya | Nigeria | Zimbabwe |
| Congo Kinshasa | Lesotho | Rwanda | |
| Cote D'Ivoire | Liberia | Senegal | |

**Table A7.** The Caribbean—5 Countries.

| |
|---|
| Guyana |
| Haiti |
| Jamaica |
| Suriname |
| Trinidad |

## Notes

[1]   As Busse and Hefeker (2007) point out, this result aligns with the findings of some studies that examine the linkages between ethnic tensions and economic growth. These studies, e.g., Easterly and Levine (1997), suggest that a high degree of conflicts attributable to racial nationality and language divisions might negatively affect economic development on average.

[2]   For instance, there are several papers on how democracy, law and order, government stability and the other political risk metrics affect FDI [e.g., Asiedu (2006); Asiedu and Lien (2011); Ali et al. (2010); Wei (2000)], yet very little has been said on how the degree of ethnocultural fractionalization influences FDI decisions of multinationals.

[3]   For example, see pages 57 to 60.

[4]   This is because people may tend to identify more with their group instead of with the society as a whole. They may be inclined to dislike other groups even if such disharmony may not tip over into an all-out civil war.

[5]   The argument is democratic institutions may have a positive impact on FDI because democracy provides checks and balances on elected officials and therefore reduces arbitrary government intervention, lowers the risk of policy reversal and strengthens property rights protection.

[6]   Li and Resnick (2003) argue that autocratic governments may be in a better position to provide more generous incentive packages and offer protection from labor unions since autocratic governments are not accountable to the citizens.

[7]   The number of forced changes in the top government; number of politically motivated assassinations or attempted assassinations of a high government official; and the number of illegal or forced changes in the ruling government.

[8]   See pages 222–25 of UNCTAD World Investment Report 2017 (https://unctad.org/webflyer/world-investment-report-2017, accessed date 4 June 2021).

[9]   Out of curiosity, we also test for non-linearity in other sub-components of the political stability index by introducing quadratic terms and interact their quadratic terms with natural resource variable, but they are insignificant.

[10]   The results on the control variables hold and the full estimation results are available from the authors upon request.

[11]   The results on the control variables hold and the full estimation results are available from the authors upon request.

[12]   Appendix A provides information on the sub-components of the political stability index and the data sources from which the index is constructed. Further information on how the index is constructed is available upon request from the authors.

[13]   Bahamas, Brunei, Hong Kong, Iceland, Iraq, North Korea, Malta, Serbia, Somalia, and Taiwan.

14  Benin, Burundi, Cambodia, Central African Republic, Chad, Djibouti, Equatorial Guinea, Georgia, Kyrgyzstan, Laos, Lesotho, Macedonia, Mauritania, Mauritius, Nepal, Rwanda, Swaziland, Tajikistan, Turkmenistan, and Uzbekistan.

15  The following 13 countries are not included in this study since they do not have the required FDI data and other control variables leaving us with 137 countries: Congo Kinshasa, Cuba, Djibouti, Dominican Republic, Ethiopia, Myanmar, Nepal, Qatar, Romania, Sudan, Syria, Turkmenistan, and Uzbekistan.

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
