# Peer review of "A Novel Measure of Political Risk and Foreign Direct Investment Inflows"

_jrfm, doi:10.3390/jrfm14100482_

Round 1
Reviewer 1 Report
In this paper authors created a new measure for political stability and investigated its relationship to FDI inflow. The topic of the paper is relevant and interesting.
Few remarks regarding results and discussion. In the description of results only low developed African countries are discussed. It is advised to authors to discuss the results for developed countries as well, and perhaps to compare the two groups. Remark regarding section 4. - it is missing the main title. Authors start with 4.1.
Author Response
- In this paper, authors created a new measure of political stability and investigated its relationship to FDI inflow. The topic of the paper is relevant and interesting. Few remarks regarding results and discussion. In description of results only low developed African countries are discussed. It is advised to authors to discuss results for developed countries as well, and perhaps to compare the two groups.
Response: We focused on that region of the world because that is the region that stands to benefit the most in terms of attracting FDI by improving its political stability scores.
- Remark regarding section 4 – it is missing the main title. Authors start with 4.1.
Response: We correct this by numbering the main title as 4. And then proceed by providing a title for 4.1, 4.2 and 4.3.
Reviewer 2 Report
Dear authors your idea is very interesting but your article in the curren form has some flaws:
In many cases there are general comments, i.e. abstract:
The proposed political risk measure contains relevant, unique and incremental information not observed in the literature. For example, although this measure is highly correlated with the political risk rating of the International Country Risk Guide (ICRG), it contains unique information that explains FDI inflows beyond what is explained by the ICRG rating.
The first link to data doesnt work.
Introduction:
Direct Investments (FDI) flows and political risk albeit with conflicting
results. Which ones??
The introduction is a mix of the results and conclusions sections, needs clarification of the structure,
Imo the introduction starts in 2nd paragraph on page 3.
There are many footnotes that make the document very difficult to read it, is recomended to analyze if they are necessary or if they can be integrated into the document.
Page 5: .
This paper contributes to this discourse by introducing a new index, the Political Stability Index, which draws inspiration from Karolyi (2015).
Its place would be in the conclusions.
Page 8:
In general, our political... Too Broad
Conclusion, include the various conclusions and recommendations spread throughout the paper.
Minor: authors use paper instead of article, study, research, sounds informal.
Author Response
1. The first link to the data does not work:
Response: We have updated the link in the paper accordingly. The dataset can be found at the following link https://www.dropbox.com/sh/hwo3rucpf5o7y50/AADTQCleBhUKizalZz8v_k0fa?dl=0
2. Direct Investment flows and political risk albeit with conflicting results. Which ones??
Response: We show in the related literature how some studies find democracy to have a positive impact on FDI (North and Weingast (1989), Li (2009)) whereas others find autocratic governments to have a positive impact on FDI (Li and Resnick (2003)). To address the author’s comment, we modify that sentence to read, “There is an extant literature that examines the relationship between Foreign Direct Investments (FDI) flows and political risk albeit with conflicting results. For example, North and Weingast (1989) and Li (2009) show that democracy has a positive effect on FDI whereas Li and Resnick (2003) argue that autocratic governments can have a positive impact on FDI.”
3. The introduction is a mix of the results and conclusion sections, needs clarification of the structure. In my opinion, introduction starts in 2nd paragraph on page 3.
Response: We respectively disagree with the referee on this point. We feel it is perfectly fine to make a brief note of the results and conclusions in the introduction section. This is especially important so that the paper does not become a “detective story” and the reader does not have to read all the way to the end to find out what has been done in the paper. Many top finance articles indeed do provide a summary of the results and conclusions as part of the introduction. Then, in the results and conclusions sections, they elaborate, as we have done in this paper.
4. There are many footnotes that make the document very difficult to read, it is recommended to analyze if they are necessary or if they can be integrated into the document.
Response: We have removed some of the footnotes and incorporated them into the article.
5. Page 5: “This paper contributes to this discourse by introducing a new index, the Political Stability Index, which draws inspiration from Karolyi (2015).” Its place would be in the conclusions.
Response: We have moved those paragraphs to the conclusions accordingly. We thank the referee for this suggestion.
6. Page 8: In general, our political … too broad
Response: We modify that statement to read: “Our political stability index and its subcomponents tend to be positively correlated with FDI and with per capita GDP and negatively correlated with the volume of natural resource exports.”
7. Conclusion: include the various conclusions and recommendations spread throughout the paper.
Response: We have modified the conclusion accordingly.
8. Authors use paper instead of article, study, research
Response: We have removed the word “paper” and incorporated the words “study” and “article” as suggested by the referee.
Reviewer 3 Report
The research should be extended beyond 2015.
The fractionalization index should be presented
Author Response
1. The research should be extended beyond 2015.
Response: We are currently unable to extend it beyond 2015 due to data limitations. We are working on an updated version of the political stability index, and we anticipate it to be available by the end of the year. Once we have it, we will be sharing on a website.
2. The fractionalization index should be presented.
Response: We do not have a fractionalization index, but we have a subcomponent of the political stability index called, Ethno-Cultural Fragility Sub-index, which amongst other inputs relies on a host of different fractionalization measures (including ethnic fractionalization, language fractionalization, religious fractionalization, and fractionalization of elites)
We have provided a link to the dataset along with a manual that details what goes into the Ethno-Cultural Fragility Sub-index:
Ethno-Cultural Fragility Sub-Index: The Ethno-Cultural Fragility component focuses on potential sources of ethnic, linguistic, and religious tensions in a country by looking at mechanisms through which ethnic, linguistic, and religious groups interact and by focusing on situations in which tensions are likely to arise. Hence, this sub-index is based on:
- Measures of ethnic, linguistic, religious, and other cultural fractionalization and polarization
- Equality of protection of rights and freedoms across social groups by the state
- The distribution and access to political power by different groups
- The fragmentation of state institutions along ethnic, class, clan, racial or religious lines, as well as brinksmanship and gridlock between ruling elites
- Divisions between different groups in society and their role in access to services or resources, and inclusion in the political process
- The extent and level of danger posed by the imbalance of power between ethnic groups
- The number of ethnic groups and the size of the dominant ethnic group, if any
The Ethno-Cultural Fragilities sub-index is generated from 18 underlying variables from 6 data sources and databases
Reviewer 4 Report
The paper is proper designed and the methodology well developed. The results are robust.
My recommendation is to take out the link from the abstract and to detail i the conclusions the benefits for the investors of the political risk.
Author Response
Thank you for your suggestion. We shall remove the link from the abstract and put it in only the appendix.
We shall modify the conclusion and explain how investors may benefit from our political risk index.
Reviewer 5 Report
The paper is well structured, with proper methodology and robust results.
I recommend to take out the link from the abstract and to extend the conclusions showing the information the investors can use in their decision.
Author Response
We shall accordingly remove the link from the abstract and extend the conclusions detailing how investors may use the our political risk index in their investment decisions.
Round 2
Reviewer 2 Report
Dear authors, I agree with your comment about the introduction in articles:
"We feel it is perfectly fine to make a brief note of the results and conclusions in the introduction section. This is especially important so that the paper does not become a “detective story” and the reader does not have to read all the way to the end to find out what has been done in the paper. "
But I consider that is not the case with your article. As a reader, for me, the detective work consists of figuring out what your research question is and why the problem is important.
For example reading paragraphs as:
Finally, using a dynamic panel data setting, Busse and Hefeker (2007) find the three most important political risk factors that matter for the investment decisions of multinationals to be the stability of the government and its ability to carry out its policies and stay in office, the strength and impartiality of the legal system and the degree of ethnic tensions. Even though it is not surprising that the degree of ethnic fractionalization affects the investment decisions of multinationals1 , we find it surprising that it is one of
the most important political risk considerations; given the paucity of coverage it has received in the FDI literature compared to the other determinants of political risk2
On the other hand, my recommendation is that authors think about whether footnotes really add to the text and contribute to its readability.